# Effects of ESMT on Microstructure and Mechanical Properties of Al-8Zn-2Mg-1.5Cu-0.15Sc-0.15Zr Cast Alloy in Squeeze Casting Process

Yongtao Xu [1,2,3,4] , Zhifeng Zhang [1,3,4,*], Purui Zhao [1,2,3,4], Zhihua Gao [1,3,4], Yuelong Bai [1,3,4,*] and Weimin Mao [2]

1    National Engineering Research Center for Non-Ferrous Metal Composites, GRINM Group, Beijing 101407, China; xyt93upc@hotmail.com (Y.X.); zpr@mail.nwpu.edu.cn (P.Z.); gaozhihua@grinm.com (Z.G.)
2    School of Materials Science and Engineering, University of Science and Technology Beijing, Beijing 100088, China; mao_wm@ustb.edu.cn
3    Grinm Metal Composites Technology Co., Ltd., Beijing 101407, China
4    General Research Institute for Nonferrous Metals, Beijing 100088, China
*    Correspondence: zhangzf@grinm.com (Z.Z.); bai_yuelong@163.com (Y.B.); Tel.: +86-1352-290-0206 (Z.Z.); +86-1381-058-5961 (Y.B.)

**Abstract:** Al-8Zn-2Mg-1.5Cu-0.15Sc-0.15Zr alloy with high-strength performance as well as good castability has been developed. In this study, effects of electromagnetic stirring melt treatment (ESMT) on microstructure and mechanical properties of the alloy in the squeeze casting process were investigated. The results show that solidification structure and mechanical properties are significantly improved by ESMT; compared with the conventional squeeze casting, the average grain size decreases from 112 μm without ESMT to 53 μm with ESMT. Meanwhile coarse primary $Al_3$(Sc, Zr) particles unavoidably occurred in cases without ESMT disappear, and segregation degree of the main elements of Zn, Mg, Cu are greatly alleviated; the tensile strength increases from 590 MPa to 610 MPa, and the elongation increases from 9% to 11%. The structure refinement and homogenization should owe to uniform temperature and composition distribution by ESMT under squeeze casting with rapid solidification.

**Keywords:** high-strength aluminum alloy; electromagnetic stirring; melt treatment; microstructure; mechanical properties; squeeze casting

## 1. Introduction

Lightweight, ultra-high-strength aluminum alloys in aerospace, rail transit, national defense, and military industries are increasingly demanded [1,2]. The castability of the cast aluminum alloys, such as Al-Zn, Al-Mg, Al-Si and Al-Cu, are fairly good, but their strength are not particularly good, which cannot satisfy the needs of high toughness. Characterized with excellent comprehensive mechanical properties processed by subsequent deformation and heat treatment, commercial Al-Zn-Mg-Cu series aluminum alloys are widely applied to substitute for low-carbon steel and cast-iron materials. However, in the alloys featured with high-alloying and large solidification temperature range, there are also problems such as coarse grains, serious segregation tendency, severe shrinkage, and high hot-tearing sensitivity by the conventional casting processes [3–5]. Much attention is being focused on squeeze casting process for this kind of alloy, because its rapid solidification under high pressure can eliminate shrinkage porosity defects in the casting process, and the casting structure becomes denser [1,6].

To achieve near-finish casting of high-strength aluminum alloy, much research has been carried out on the forming process and composition design in recent years [7–9]. On one hand, casting defects in the small-sized squeeze castings can be reduced to some degree for some wrought Al-Zn-Mg-Cu alloys, but there are still unavoidable defects

such as shrinkage porosity, and coarse structure in the large-sized squeeze castings, which seriously affect casting performance and mechanical properties [10,11]. In response to this problem, an advanced electromagnetic stirring melt treatment (ESMT) method was developed [12,13], and uniform, fine solidification structure of wrought 7XXX series alloys was obtained by applying electromagnetic stirring in the solidification process [14–16]. On the other hand, an ultra-high-strength cast aluminum alloy for squeeze casting, designed by main element composition optimization and Sc and Zr micro-alloying, exhibited good casting properties while maintaining mechanical properties with high strength and high toughness [17]. However, little work has been done to examine the effect of the ESMT on the microstructure and mechanical properties of the cast alloy.

The aim of this work is to study the solidification characterization of the new Al-8Zn-2Mg-1.5Cu-0.15Sc-0.15Zr cast alloy, where effects of ESMT on microstructure, composition, and mechanical properties of squeeze castings are explored, and also corresponding mechanisms of structure refinement and homogenization are discussed.

## 2. Materials and Methods

The industrial high-purity aluminum (99.99%), pure zinc (99.92%), pure magnesium (99.95%), Al-50Cu, Al-5Zr, and Al-2Sc master alloys were adopted to prepare the experimental alloy, and its practical chemical composition is shown in Table 1. The liquidus temperature is calculated to 632 °C by the JMatPro software. Figure 1 shows the schematic view of the ESMT apparatus. The test alloy was first heated to 760 °C, and then the alloy melt was poured into a stainless-steel crucible with a size of φ70 mm × H180 mm and a pre-heating temperature of 350 °C. In the case with ESMT, as the melt temperature is reduced to 660 °C, the melt was poured into a cylindrical die with a size of φ60 mm × H90 mm and a preheating temperature of 150 °C to squeeze castings, and the melt processing parameters and forming pressures of the die forming machine are shown in Table 2. In the case without ESMT, the melt was naturally cooled to about 660 °C. The detailed information about experimental process procedures is provided in literature [12]. To obtain the temperature change of the alloy melt in the experiments under two conditions, the temperature curve with time at different positions in the melt were measured, and the testing positions of the thermocouples are shown in Figure 1, named 1, 2, 3, 4, 5 and 6.

**Table 1.** Practical chemical composition of the test alloy (wt. %).

| Zn | Mg | Cu | Zr | Sc | Fe | Si | Others | Al |
|------|------|------|------|------|------|------|--------|------|
| 7.94 | 1.94 | 1.48 | 0.14 | 0.14 | 0.02 | 0.02 | ≤0.01 | Bal. |

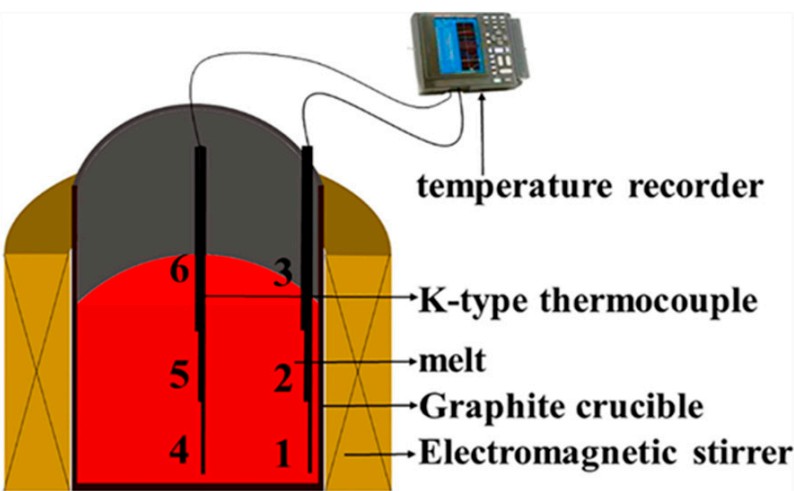

**Figure 1.** Schematic diagram of ESMT.

**Table 2.** Melt treatment parameters under different conditions.

| Conditions | Pouring Temperature/°C | Stirring Frequency/Hz | Stirring Current/A | End Temperature/°C | Ratio PRESSURE/MPa |
|---|---|---|---|---|---|
| No ESMT | 760 | 0 | 0 | 632 | 100 |
| ESMT | 760 | 5 | 10 | 632 | 100 |

Metallographic samples with a size of 25 mm × 10 mm × 6 mm were cut along the radial direction of the squeeze castings. The specimens were grounded, polished, and then anodized in a diluted solution of 2.5%$HBF_4$ acid. The coating voltage is 30 V, the coating current is controlled below 1 mA, and the coating time is 60 s. The microstructures were observed by ZEISS optical microscope and the grain size was evaluated using the linear intercept method described in ASTM standard E112-96. The solid solution level of elemental line scan analysis and secondary phases were observed on a JSM-7900F scanning electron microscope. Element distribution in squeeze castings was measured by direct reading spectrometer, by measuring three points at each location and calculating the average. The tensile testing samples was subjected to T6 heat treatment [17] for the mechanical properties measurement according to GB/T228.11-2010, by taking three samples along the axis of each casting for measuring at least 3 times to ensure the stability of the experimental results. Tensile properties were performed on a CSS-44100 electronic universal testing machine with a 2 mm/min loading speed. The tensile specimens are illustrated in Figure 2.

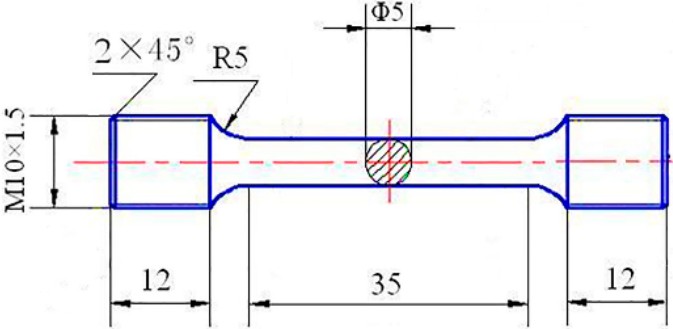

**Figure 2.** Sample geometry of a tensile specimen (unit: mm).

### 3. Results

*3.1. Microstructure Characterization*

Figure 3 shows the optical microstructures of squeeze castings of the test alloy under two conditions: (a) without ESMT; (b) with ESMT. It can be seen from Figure 3a that the alloy melt without ESMT consists of coarse dendritic grains with irregular morphology, and the average grain size is 112 μm by the cross-section method. In the case with ESMT shown in Figure 3b, the microstructure is significantly refined, and the grains are mainly made of equiaxed spherical crystal structures with an average grain size of 53 μm. It is clear that the grain size and microscopic morphology are improved greatly when the ESMT is applied to the test alloy melt during squeeze casting process.

Figure 4 shows the scanning images of the alloy microstructure of the squeeze castings under different conditions. It can be seen that there are rectangular and triangular bulk primary second phases inside the grains without ESMT, and the size of the primary second phase is about 10 μm, as shown in the black circle in Figure 4a, which is $Al_3$(Sc, Zr) primary phase by EDS analysis, as shown in Figure 5. After ESMT, the coarse primary $Al_3$(Sc, Zr) phases inside the grains decrease or even nearly disappear, as shown in Figure 4b.

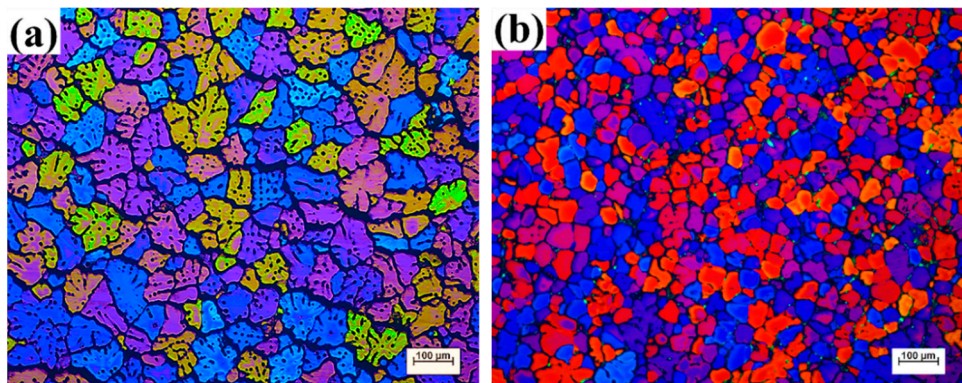

**Figure 3.** Optical microstructures of squeeze castings of the test alloy under two conditions: (**a**) Without ESMT; (**b**) With ESMT.

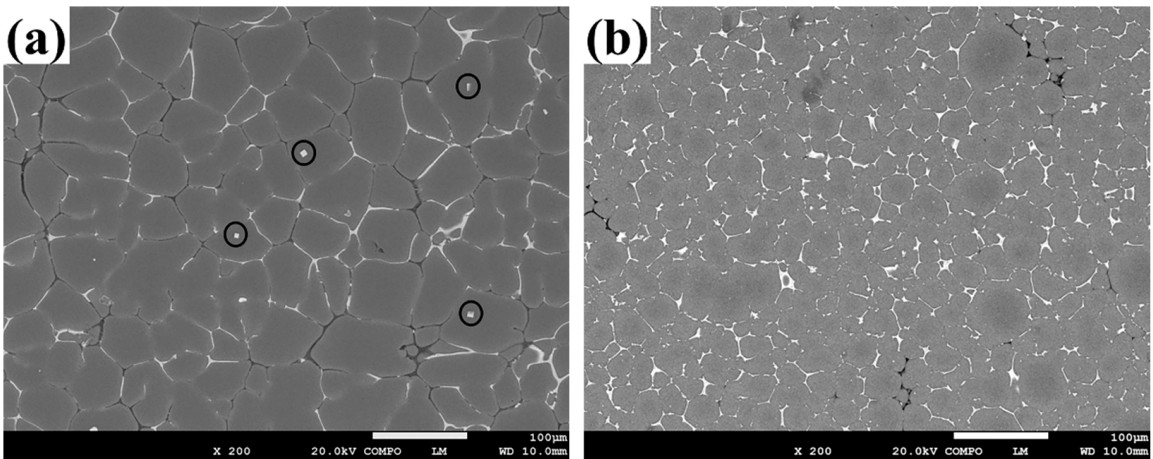

**Figure 4.** SEM microstructures of squeeze castings of the test alloy under two conditions: (**a**) Without ESMT; (**b**) With ESMT.

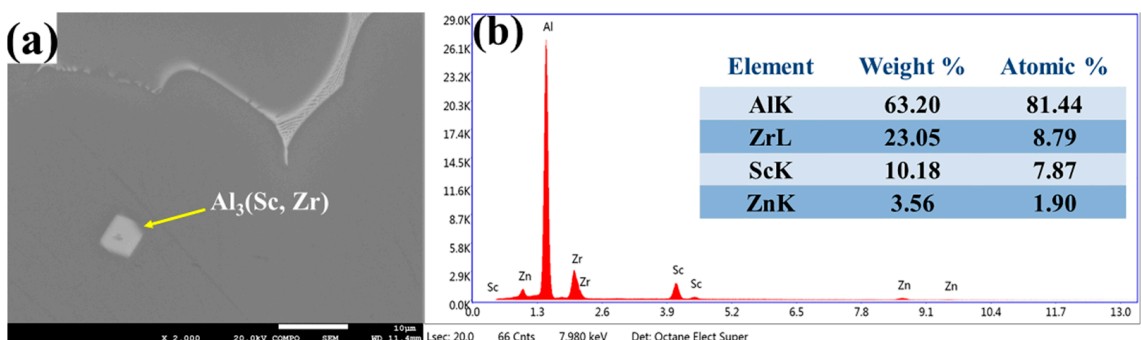

**Figure 5.** Coarse bulk primary $Al_3(Sc, Zr)$ phase (**a**), and EDS analysis result (**b**) in SEM microstructures of squeeze castings without ESMT.

### 3.2. Composition Analysis

Figure 6 shows the concentration distribution of Zn, Mg, and Cu elements in different positions of the castings formed by squeeze casting under different treatment conditions. It is noted that, the concentration distribution of alloying elements has a decreasing trend from the center to the edge in the castings without ESMT, and the difference in the distribution of element concentration is very large. After the ESMT, the difference in the distribution of element concentrations in the castings decreases obviously. It is clear that the ESMT can reduce the macro-segregation degree of the alloying element concentration.

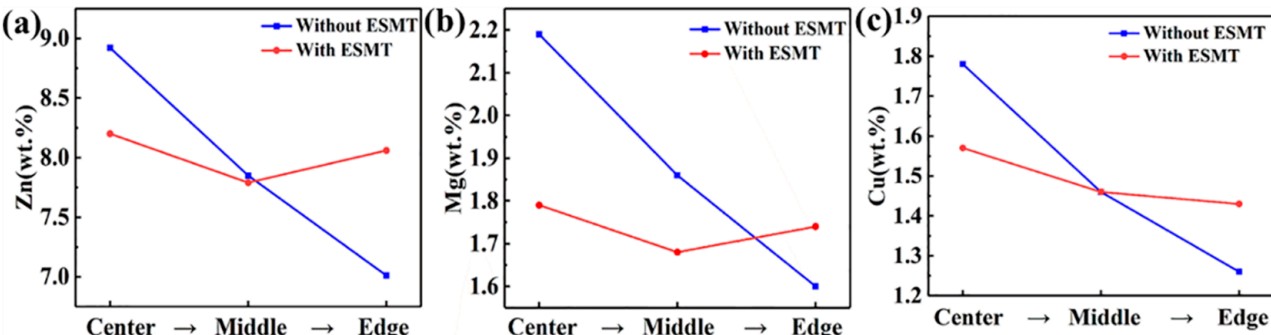

**Figure 6.** Concentration distribution of Zn, Mg, and Cu elements in different positions of the squeeze castings under different treatment conditions: (**a**) Zn; (**b**) Mg; (**c**) Cu.

Figure 7 shows the line-scan element distribution among grains at different positions of squeeze castings under different processing conditions. In the case without ESMT, shown in Figure 7a,b, the degree of element segregation at the edge is more serious than at the center, and the segregation position of Zn, Mg, and Cu elements exist at the grain boundary in the microstructure. In the case with ESMT, shown in Figure 7c,d, the degree of element segregation between the edge and the center decreases, especially at the grain boundary in the microstructure. It is obvious that the ESMT can improve the composition homogenization of the castings, as well as at the grain boundary.

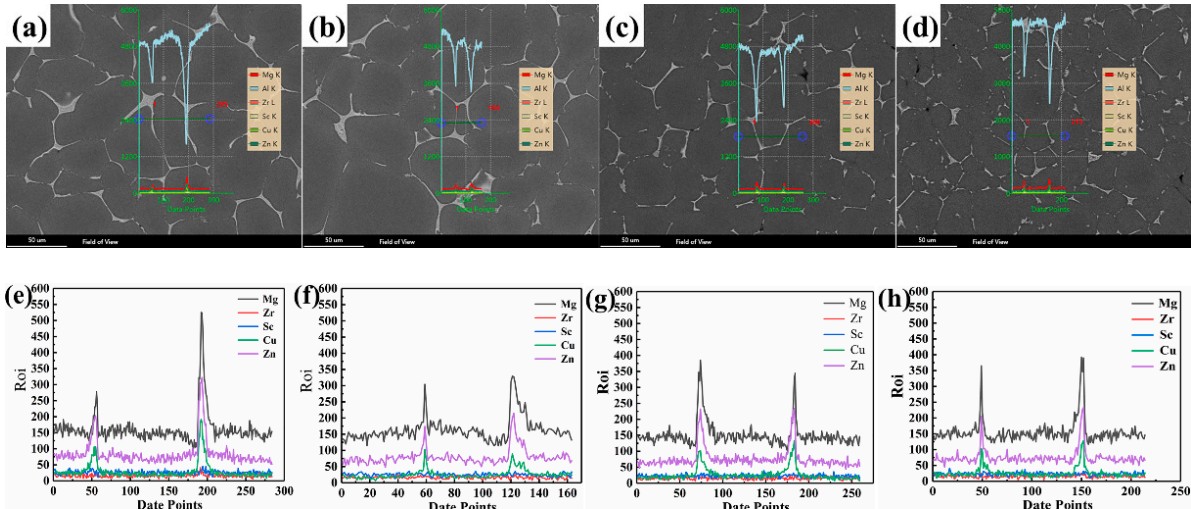

**Figure 7.** Line scanning element distribution among grains at different positions of squeeze castings under different conditions: (**a**,**e**) Without ESMT, edge; (**b**,**f**) Without ESMT, center; (**c**,**g**) With ESMT, edge; (**d**,**h**) With ESMT, center.

### 3.3. Mechanical Properties

Figure 8 shows the mechanical properties of squeeze castings under different conditions. It can be seen from the figure that the mechanical properties of the alloy are improved by ESMT. The tensile strength increases from 590 MPa without ESMT to 610 MPa with ESMT, and the elongation increases from 9% without ESMT to 11% with ESMT. Furthermore, the mechanical property error reduces greatly with ESMT, and the performance of the alloy is more stable.

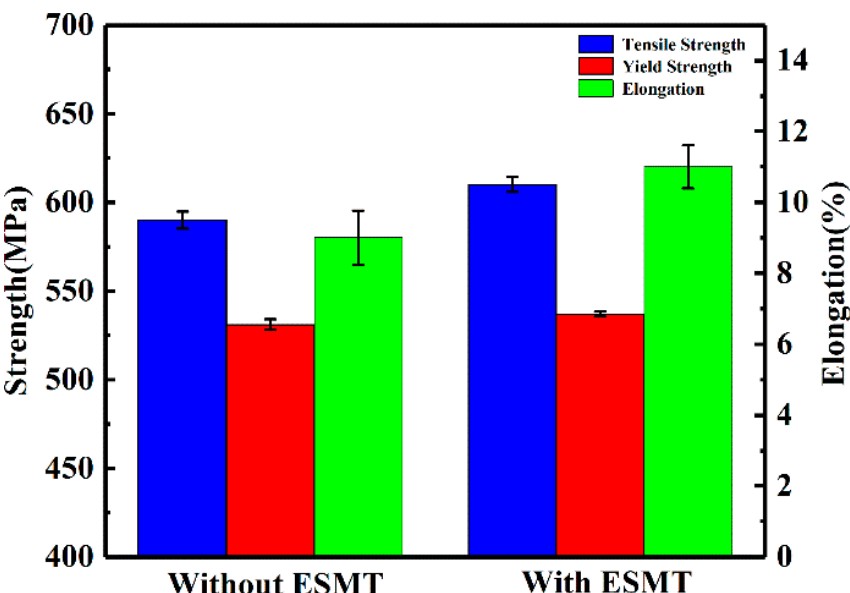

**Figure 8.** Mechanical properties of squeeze castings under different conditions.

Figure 9 shows the fracture surfaces of squeeze castings under different conditions: (a) without ESMT; (b) with ESMT. It can be seen from Figure 9a that the tensile fracture is mostly composed of shear planes and a few small dimples, indicating the main fracture mode of intergranular shear fracture. Figure 9b shows the tensile fracture morphology when the alloy is treated with ESMT. The fracture morphology of the alloy is basically the same as the alloy without ESMT, but the number of dimples increased, especially the intergranular dimples which increased substantially. In addition, there will be more transgranular dimples on the fracture surface.

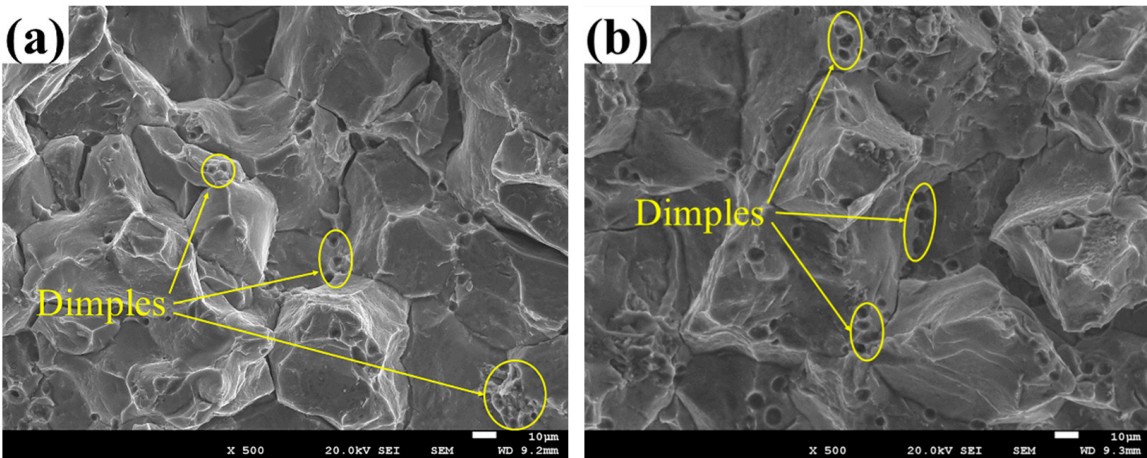

**Figure 9.** The tensile fracture morphology of squeeze castings under different conditions: (**a**) Without ESMT; (**b**) With ESMT.

## 4. Discussion

Al-8Zn-2Mg-1.5Cu-0.15Sc-0.15Zr cast alloy, designed by main element composition optimization and Sc and Zr microalloying, exhibited good casting properties while maintaining mechanical properties with high strength and high toughness [17]. However, the alloy is featured with high-alloying, grain refinement, and homogenization which have a great effect on mechanical properties of the castings.

For the untreated melt, since there is only natural convection in the melt, the cooling of the melt mainly relies on the heat conduction with the crucible wall for heat dissipation. The

stainless-steel crucible wall dissipates heat faster, so the melt temperature near the crucible wall is relatively low. The melt away from the crucible wall has a slow heat conduction speed, leading to a large temperature gradient and uneven temperature distribution as shown in Figure 10a. When ESMT is applied, forced convection is caused by the electromagnetic force driving the melt flow, and the temperature distribution is more uniform as shown in Figure 10b. During the solidification process of the melt without ESMT, the temperature field and composition field of the melt is not uniformly distributed, and the Sc-Zr atomic clusters have a high aggregation ability, which leads to the aggregation and segregation of Sc and Zr elements, and the $Al_3(Sc, Zr)$ phase in the melt first nucleates and grows due to the higher liquidus temperature as shown in Figure 11a, and subsequently coarse $Al_3(Sc, Zr)$ particles cannot be used as a nucleation base to refine the microstructure of the alloy, which will eventually lead to the coarse and uneven grains in the castings.

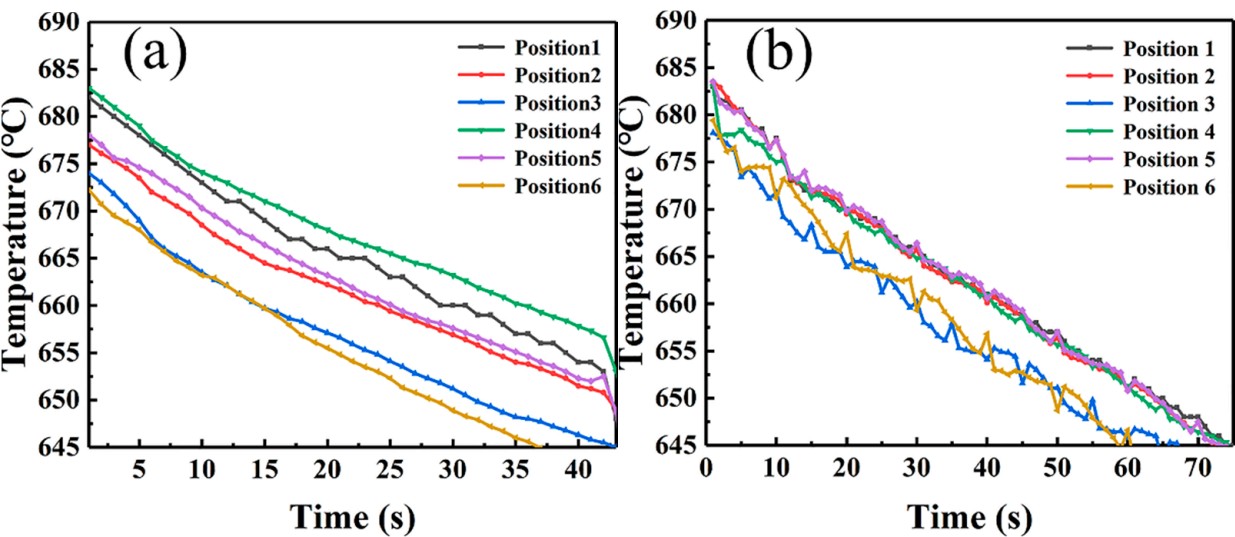

**Figure 10.** Cooling curves of temperature with time under different conditions: (**a**) Without ESMT; (**b**) With ESMT.

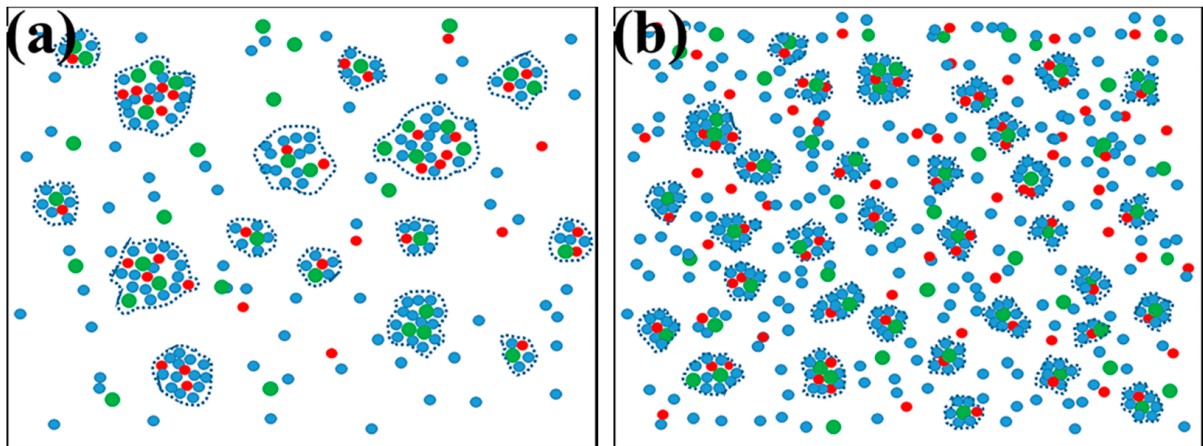

**Figure 11.** Schematic diagram of Al3(Sc, Zr) nucleation: (**a**) Without ESMT; (**b**) With ESMT; ● Al atom group, ● Sc atom group, ● Zr atom group.

When applying ESMT to the melt, the temperature field and composition field of the melt is uniformly distributed, and Sc and Zr elements are more evenly distributed in the melt, and the trend of aggregation and segregation of Sc and Zr elements is alleviated, and lots of $Al_3(Sc, Zr)$ phases in the melt simultaneously nucleate as the melt temperature drops to the liquidus, as shown in Figure 11b; consequently, a large number of fine and dispersed

primary $Al_3$(Sc, Zr) crystal nuclei have not enough time and space to grow, and the tiny $Al_3$(Sc, Zr) particles can be used as effective nucleation bases to refine the microstructure of the alloy, which will eventually lead to the fine and even grains in the castings.

At the same time, the solute concentration in the liquid phase can be homogenized by strong shearing action of ESMT, and the uneven distribution of solute elements in the case without ESMT can be improved as well. According to the Hall-Petch formula, the mechanical properties of the alloy increase with the decrease of the grain size, so the mechanical properties of the alloy castings by ESMT are improved. It is emphasized in this study that fine microstructure and homogenous composition distribution will contribute to elongation enhancement.

## 5. Conclusions

In this study, effects of ESMT on microstructure and mechanical properties of Al-8Zn-2Mg-1.5Cu-0.15Sc-0.15Zr cast alloy in the squeeze casting process were investigated. The following conclusions can be made:

(1) Solidification structure and mechanical properties are significantly improved by ESMT; compared with the conventional squeeze casting, the average grain size decreases from 112 μm without ESMT to 53 μm with ESMT, and segregation degree of the main elements of Zn, Mg, Cu are greatly alleviated.

(2) Rectangular and triangular bulk primary second phases with a size of about 10 μm inside the grains unavoidably occurred in cases without ESMT decrease or even nearly disappear by ESMT.

(3) The tensile strength increases from 590 MPa without ESMT to 610 MPa with ESMT, and the elongation increases from 9% without ESMT to 11% with ESMT. The improvement of the mechanical properties should owe to structure refinement and composition homogenization by ESMT under squeeze casting with rapid solidification.

**Author Contributions:** Z.Z., Z.G., and Y.B. devised the project. Z.Z. supervised the work and reviewed the manuscript. Y.X., and P.Z. carried out experiments, simulations, and wrote the manuscript. W.M. put forward the conceptualization and methodology. All authors have read and agreed to the published version of the manuscript.

**Funding:** This research received no external funding.

**Institutional Review Board Statement:** Not applicable.

**Informed Consent Statement:** Not applicable.

**Data Availability Statement:** Date is contained within the article.

**Acknowledgments:** The work support by GRINM Metal Composites Technology Co., Ltd. (Enterprise Technology Re-search Project) and research platform support by National Engineering and Technology Research Center for Non-ferrous Metal Matrix Composites, GRINM Group Co., Ltd. is greatly acknowledged.

**Conflicts of Interest:** The authors declare that the research was conducted in the absence of any commercial or financial relationships that could be construed as a potential conflict of interest.

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
