# Peer review of "Effects of ESMT on Microstructure and Mechanical Properties of Al-8Zn-2Mg-1.5Cu-0.15Sc-0.15Zr Cast Alloy in Squeeze Casting Process"

_crystals, doi:10.3390/cryst12070996_

Round 1

Reviewer 1 Report

Interesting article but requires supplementing and explanations.

I have some comments on the results. The increase in strength from 590 MPa to 610 MPa is an increase of 3.5%, a value within the statistical error. There is visible grain reduction after the ESMT process, but a slight increase in strength.

1) How many samples have been tested for strength? Three pieces of melt were cast?

2) How was the strength test specimen prepared?

3) What do the numbers 1, 2, 3, 4, 5, 6 mean in fig. 1

Editing notes:

In Fig. 3, replace the red circles with black ones.

Author Response

Response to Reviewer 1 Comments

Dear reviewer:

For the age hardening alloy, grain size is not the main strengthening mechanism. Therefore, the grain size is refined, but the increase in the mechanical properties of the alloy is not very obvious.

Point 1: How many samples have been tested for strength? Three pieces of melt were cast?

Response 1: Three tensile samples were taken from each alloy castings.

Point 2: How was the strength test specimen prepared?

Response 2: Three tensile samples were taken radially from each alloy ingot.

Point 3: What do the numbers 1, 2, 3, 4, 5, 6 mean in fig. 1

Response 3: The numbers 1, 2, 3, 4, 5, 6 mean six different positions in figure 1.

Point 4: Editing notes:

In Fig. 3, replace the red circles with black ones.

Response 4: I have changed it.

Reviewer 2 Report

This work studied the effect of ESTM on the evolution of microstructure and mechanical properties in squeeze-casted Al-Zn-Mg-Cu-SC-Zr alloy. With ESTM, interesting results are presented, such as the refined grain structure, reduced segregation level and improved mechanical properties. However, there are some points to be clarified as below: 

(1). In the introduction part, it should make it clear that why new Al-Zn-Mg-Cu-Sc-Zr alloy is studied in this work. 

(2). Though the grain size in Fig. 2 after ESMT is quite fine than that without ESMT, however, it seems the observations in Fig. 3 are not supporting this since the grain structure in Fig. 3 are quite similar for both. Meanwhile, the grain size calculated from Fig. 6 is neither support the finer grain size after ESMT. For instance, this should be the grain size between two peaks but you will find the grain size are kind of similar with/without ESMT. 

(3). How many points are measured to get the solid solution level of elements in Fig. 5? STD should be added for this. 

(4). What's "Roi" standing for in Fig. 6? 

(5). From Fig. 7, it can be mentioned that the mechanical properties are improved after ESMT but the extent is very small. However, author mentioned in Fig. 8 that the fracture surfaces are totally different, from trans-granular shear to trans-granular dimples, which is hard to believe. Meanwhile, these fractures seem to be more inter-granular rather than trans-granular. Author should pay more attention on the fracture analysis. 

Author Response

Response to Reviewer 2 Comments

Dear reviewer:

Thank you for your comments, I have made changes based on your comments.

Point 1: In the introduction part, it should make it clear that why new Al-Zn-Mg-Cu-Sc-Zr alloy is studied in this work.

 Response 1: The castability of the cast aluminum alloys, such as Al-Zn, Al-Mg, Al-Si and Al-Cu, are pretty good, but theirs strength are not particularly good, which cannot satisfy the needs of high toughness.

Point 2: Though the grain size in Fig. 2 after ESMT is quite fine than that without ESMT, however, it seems the observations in Fig. 3 are not supporting this since the grain structure in Fig. 3 are quite similar for both. Meanwhile, the grain size calculated from Fig. 6 is neither support the finer grain size after ESMT. For instance, this should be the grain size between two peaks but you will find the grain size are kind of similar with/without ESMT.

Response 2: The picture selected in Figure 4(b) is the structure of the alloy at different positions, and the grain size will be different. I did not notice the grain size of the two groups of alloys. The line scan of a single grain in Figure 7 is not a radial scan analysis of spherical crystals, and does not represent the grain size of a single grain

Point 3: How many points are measured to get the solid solution level of elements in Fig. 5? STD should be added for this.

Response 3: I measured three points at each location and calculated the average.

Point 4: What's "Roi" standing for in Fig. 6?

Response 4: Roi reflects the element content in the alloy.

Point 5: From Fig. 7, it can be mentioned that the mechanical properties are improved after ESMT but the extent is very small. However, author mentioned in Fig. 8 that the fracture surfaces are totally different, from trans-granular shear to trans-granular dimples, which is hard to believe. Meanwhile, these fractures seem to be more inter-granular rather than trans-granular. Author should pay more attention on the fracture analysis.

Response 5: The improvement of the mechanical properties of the alloy is relatively small, and the number of dimples on the fracture surface increases after ESTM.

Reviewer 3 Report

The authors in their work have shown the positive effect of electromagnetic stirring melt treatment on the microstructure and mechanical properties of the Al-8Zn-2Mg-1.5Cu-0.15Sc-0.15Zr alloy. This paper is well written and can be suitable for publication after some revision. There are several comments on the article.

1) Introduction needs a significant improvement. Thus, there are a number of works devoted to the electromagnetic stirring melt treatment of Al-Zn-Mg-Cu alloys, which should be refererred to. For example: Zhang ZF, Wang ZG, Li B, Xu J, Wang R. Effect of A-A-EMS Melt Treatment Process on Microstructure and Property of Squeeze Casting Al-Zn-Mg-Cu-Sc Alloys. Materials Science Forum. 2013; 765: 321-325, Jun Xu, Zhi Hua Gao, Zhi Feng Zhang, Men Gou Tang, Wei Dong Yu. Application Research on Annular Electromagnetic Stirring Casting Process of Al-Zn-Mg-Cu Alloy. Solid State Phenomena. 2012; 192-193: 466-469 and many others.

2) The testing machine for measurement of mechanical properties should be specified.

3) It is unclear why the authors call their casting process without pressure application as squeeze casting. It should be gravity casting. Otherwise, it is necessary to describe the casting technique in more detail, indicating the applied pressure.

4) In the Materials and Methods section, the Authors write that, in the first case, the test alloy "was poured into a stainless-steel crucible with a size of φ70mm×H180mm and a preheating temperature of 350°C" and, in the second case, "the melt was poured into a cylindrical die with a size of φ60mm×H90mm and a preheating temperature of 150°C". It is not clear, because in these two cases there are different cooling rates the temperature distribution.

Author Response

Response to Reviewer 3 Comments

Dear reviewer:

Thank you for your comments, I have made changes based on your comments.

Point 1: Introduction needs a significant improvement. Thus, there are a number of works devoted to the electromagnetic stirring melt treatment of Al-Zn-Mg-Cu alloys, which should be refererred to. For example: Zhang ZF, Wang ZG, Li B, Xu J, Wang R. Effect of A-A-EMS Melt Treatment Process on Microstructure and Property of Squeeze Casting Al-Zn-Mg-Cu-Sc Alloys. Materials Science Forum. 2013; 765: 321-325, Jun Xu, Zhi Hua Gao, Zhi Feng Zhang, Men Gou Tang, Wei Dong Yu. Application Research on Annular Electromagnetic Stirring Casting Process of Al-Zn-Mg-Cu Alloy. Solid State Phenomena. 2012; 192-193: 466-469 and many others.

 Response 1: I have made changes based on the comments made by the reviewers.

Point 2: The testing machine for measurement of mechanical properties should be specified.

Response 2: Tensile properties were performed on a CSS-44100 electronic universal testing machine with a 2mm/min loading speed.

Point 3: It is unclear why the authors call their casting process without pressure application as squeeze casting. It should be gravity casting. Otherwise, it is necessary to describe the casting technique in more detail, indicating the applied pressure.

Response 3: The process method in the experiment is squeeze casting. The pressure parameters in the experiment are listed in table 2.

Point 4: In the Materials and Methods section, the Authors write that, in the first case, the test alloy "was poured into a stainless-steel crucible with a size of φ70mm×H180mm and a preheating temperature of 350°C" and, in the second case, "the melt was poured into a cylindrical die with a size of φ60mm×H90mm and a preheating temperature of 150°C". It is not clear, because in these two cases there are different cooling rates the temperature distribution.

Response 4: The alloy melt was stirred and cooled in a crucible, and after reaching a specific temperature, it was cast into a mold and formed under pressure. The parameters of the two melts were the same throughout the experiment.

Round 2

Reviewer 2 Report

The modifications are satisfied and then it can be suggested to accept.